# Stakeholder perspectives on a patient-centred intervention (DIALOG+) for adolescents with common mental disorders in Colombia: A qualitative study

**Carlos Gómez-Restrepo**[1], **Arturo Marroquín-Rivera**[2], **María Gabriela Calvo-Valderrama**[2], **Laura Ospina-Pinillos**[3], **Diliniya Stanislaus Sureshkumar**[4], **Victoria Jane Bird**[4]*

1 Departments of Clinical Epidemiology and Biostatistics, and Psychiatry and Mental Health, Faculty of Medicine, Pontificia Universidad Javeriana, Hospital Universitario San Ignacio, Bogotà D.C, Colombia, 2 Department of Clinical Epidemiology and Biostatistics, Pontificia Universidad Javeriana, Bogotà D.C, Colombia, 3 Department of Psychiatry and Mental Health, Faculty of Medicine, Pontificia Universidad Javeriana, Bogotà D.C, Colombia, 4 Unit for Social and Community Psychiatry, Wolfson Institute of Population Health, Queen Mary University of London, London, United Kingdom

* v.j.bird@qmul.ac.uk

**Data Availability Statement:** Data cannot be shared publicly because of confidentiality issues as

## Abstract

### Background

Patient and stakeholders' involvement in the development of mental health interventions is a central part of the research process as end-user's input can improve the design of patient-centered interventions. This is particularly important when developing interventions directed towards improving the mental health of children and adolescents. The rising prevalence of mental health disorders in this population requires special attention and the development of interventions that include them as active participants is crucial.

### Objective

Our aim is to explore the perspectives and opinions of adolescents, parents, educators/youth workers, and clinicians regarding the appeal and usability of an existing patient-centered digital intervention (DIALOG+), which aims to improve quality of life.

### Methods

As part of a broader study aiming to adapt and test DIALOG+, we conducted Online Focus Groups (OFGs) with adults and adolescents in two cities in Colombia. The existing DIALOG+ intervention was introduced to participants, followed by a structured discussion regarding the opinions and views of stakeholders. A framework approach was used to identify the main themes followed by content analysis to aid adaptation.

### Results

We conducted 10 OFGs with a total of 45 participants. A positive feature highlighted by all groups was the innovation of including a digital intervention in a traditional medical visit.

the raw data are verbatim transcripts, as per the requirements of comité de Investigaciones y Ética Institucional, Facultad de Medicina, Pontificia Universidad Javeriana. Data are available on request from the chair of the International Steering Committee (Professor Domenico Giacco) to researchers who meet the criteria for access to confidential data. As the dataset relates to interviews transcribed verbatim anyone wishing to access the original qualitative dataset, is required to contact the International Steering Committee chair (domenico.giacco@warwick.ac.uk). A formal written request outlining the nature of the project is required. Permission to utilise the data would then be sought from the other members of the study's independent advisory board, who provide oversight on all ethical issues relating to the study and liaise directly with the relevant ethical committees.

**Funding:** All authors are supported by the Medical Research Council (grant number: MR/S023674/1) Building resilience in adolescence - improving quality of life for adolescents with mental health problems in Colombia (BRiCs study). https://mrc.ukri.org/ The funders had no role in study design, data collection and analysis, decision to publish, or preparation of the manuscript.

**Competing interests:** The authors have declared that no competing interests exist.

Additionally, participants considered that the active role that adolescents have when using the intervention empowers them. Barriers identified included concerns from clinicians related to the time required during consultations and confusion with terminology. Furthermore, additional domains that are particularly relevant for the adolescent population were suggested.

## Conclusions

Data obtained suggest that overall, the DIALOG+ intervention and supporting app are seen as innovative and appealing to adolescents as well as adult stakeholders. However, concerns raised about the availability of time to apply the intervention, the app interface and the language and terminology require modification.

## Introduction

Mental health disorders are one of the leading causes of burden of disease worldwide and their prevalence is rising. In 2016, it was estimated that they affected more than 1 billion people and were responsible for 19% of all years lived with disability [1]. More concerning, is their impact on the youth; globally, it is estimated that approximately 10–20% of children and adolescents are affected by mental health disorders [2, 3]. This may be due to the intrinsic vulnerability of children and adolescents to mental health problems resulting from key biological, psychological and social changes that affect individuals during this period [4]. Despite their prevalence, the mental health needs of children and adolescents are often overlooked, particularly in Low and Middle-Income Countries (LMICs) [2].

Colombia is an upper middle-income country with a history of armed conflicts, forced internal displacement and violence that has significantly impacted the lives of its population [5]. Consequently, some attempts have been made to give special attention to the mental health needs of Colombian adolescents in recent years [6, 7], however the impact of this endeavor is poorly studied. According to the National Mental Health Survey (ENSM-2015) the prevalence of any mental health disorders in adolescents between 12 and 17 years old was 7.2%, with anxiety and affective disorders being the most frequent [8].

To help address the considerable impact of mental health disorders, different interventions have been developed which aim to improve a range of health outcomes. Patient involvement in the development of interventions has gradually gained importance in research and in clinical practice [9], especially due to evidence suggesting that patient-centered interventions may improve quality of life and enhance adherence to treatments [10, 11]. Although, most studies involve adult individuals, some studies suggest that patient-centered interventions in youth populations -allowing active involvement in the decisions of their health care- improve symptoms and provide greater overall satisfaction too [12, 13]. Shared decision-making may increase adolescent engagement because with this approach, the patient and clinician work together as a team and trust is improved [14]. Additionally, the use of technology can make interventions that are aimed at young people more appealing due to greater access and more engaging and interactive content. Furthermore, the use of certain mental health intervention applications (or apps) has been positively received by adolescents, carers and clinicians in some settings [15].

The aim of this study is to explore the views of adolescents, parents, teachers, and clinicians regarding an existing patient-centered digital intervention called DIALOG+, as well as our

experience involving young people as an active part of our research. DIALOG+ is a patient-centered intervention mediated by an app that focuses on structuring communication between clinicians and patients, offering a solution-focused approach [10, 16, 17]. The use of this intervention has evidence of effectiveness in adults in the United Kingdom, where improved Subjective Quality of Life scores, psychopathological symptoms and social outcomes were evidenced at follow up [16, 18]. Some of the strengths that the DIALOG+ intervention offers, in addition to a straight forward interface, is the structured nature that it gives to a routine consultation as well as the focus that it gives to the patients self-reported needs. This, added to the positive experience with adult populations, makes it a strong intervention to test with a different patient group like adolescents and in a different context like Colombia.

The intervention assesses 11 key life topics or domains, eight life-related (mental health, physical health, job situation, accommodation, leisure activities, partner and family, friendships, personal safety) and three treatment-related, (medications, practical help, and meetings with mental health professionals). The clinician asks the patient to rate their satisfaction in each of these domains on a scale of 1 to 7, where 1 is '*completely unsatisfied*' and 7 '*completely satisfied*'. Domains and their descriptions can be found in S1 Table in S1 File [18]. After each question, the patient is asked if they require any additional support in that particular area of their life. This gives the clinician valuable information on the priorities and/or preferences of the patient. Following a review of the 11 items, up to three are then jointly selected by the patient and clinician. A 4-step approach, based on the principles of solution-focused therapy is used to address the patient's concerns and agree on further actions. with a four-step approach based on brief solution focused therapy used to address the patients concerns. The process allows the clinician to give tailored assignments that target specific domains that are in need of improvement, which are then discussed in future consultations.

This paper focuses on the initial impressions of the DIALOG+ intervention, as a new intervention to improve adolescent mental health in this context. Specifically, it will indicate which aspects of the intervention do and do not appeal to adolescent users and other stakeholders. Investigating stakeholders' views on the initial appeal and acceptability of the intervention will assist with adaptation and implementation.

## Methods

This qualitative study was conducted within the context of a larger project called "Building Resilience in Adolescence -improving quality of life for adolescents with mental health problems in Colombia" (BRiCs) which aims to improve outcomes for adolescents with depression and/or anxiety in Colombia. The project was a collaborative research project between Queen Mary University of London, UK and the Pontificia Universidad Javeriana (PUJ) in Bogotá.

In order to adapt this intervention to the Colombian adolescent population and their context, we conducted Online Focus Groups (OFGs). This allowed us to obtain the perspectives of users, key stakeholders and gatekeepers (parents/guardians, youth workers and educators) regarding the appeal and usability of the intervention.

The OFGs were conducted between June and November of 2020 and were initially planned to take place as traditional face-to-face focus groups. However, the sudden surge of the COVID-19 pandemic caused a state of emergency in Colombia in March 2020 and subsequent lockdown and physical distancing measures were imposed. In order to continue the research, an ethics amendment was requested to change the Focus Group (FG) methodology from face-to-face to synchronous OFGs through a videoconferencing platform (Microsoft teams®). Changes to the research protocol were approved by the Institutional Review Board (IRB) of both academic institutions and clinical settings (Protocol FM-CIE-0084-20).

## Participants

Participants were recruited through a convenience sampling method from two clinical settings in Colombia: Bogotá (Colombia's capital city) and Duitama (Intermediate city).

The inclusion criteria for adolescents were: i) aged between 13 and 16 years of age, ii) current or previous self-reported symptoms of depression and/or anxiety, iii) a willingness to share their experience in an OFG and iv) capacity to provide informed assent. Parents or guardians were included, if they provided care to an adolescent aged between 13 and 16 years-old with current or previous self-reported anxiety and/or depression and had the capacity to provide informed consent for their child to participate in the study. Finally, clinicians, educators, and youth workers were i) required to have experience working with adolescents experiencing depression and/or anxiety ii) be aged 18 years old or over. Once participants agreed to take part in the study, a sociodemographic questionnaire was sent to their e-mails and they were asked to complete it and send it to the study coordinator before the OFG session. The groups had between 3 to 7 participants and were scheduled for a duration of 120 minutes. Participants were reimbursed for their time with a $40.000 COP (approx. 12 USD) voucher.

## Procedure

Before any data collection began, written signed informed consent was obtained. This included an electronic signature, as it was not possible to obtain ink signatures due to the COVID-19 pandemic. Where individuals were under 18, written informed assent from participants and their parents/guardians was obtained. Researchers made sure that participants understood that they could withdraw at any given moment of research without any consequences as well as ensuring the anonymity and security of data obtained. Once consent was obtained, the study coordinator sent two invitations for two different meetings. The first meeting included a trial run to check the participant's internet connection and to confirm that all the participants were able to join the OFG and use the videoconferencing software without difficulties. Additionally, during this first trial meeting the coordinator explained further details of the project and solved logistic and participation queries. The second invitation was for the OFG session. For both invitations the time, date and agenda were included.

The sessions were facilitated by two of the teams' principal investigators (psychiatrists and researchers LOP and CGR) and an anthropologist, all with extensive experience conducting FGs. Throughout the sessions, facilitators were aware of maintaining a moderate speech speed and avoided the use of technical or confusing terminology in order to propitiate familiarity and encourage participation.

The sessions included the following steps: 1) an introduction of the facilitator and of each participant with an explanation of the dynamics of the OFG 2) Verifying understanding of the informed consent 3) Introduction and description of the DIALOG+ intervention 4) Sharing the DIALOG+ app with participants, explaining its features and its aim 5) a structured discussion about the intervention and its content, clarity, usefulness, design and overall thoughts using a topic guide 5) a final summary. A general topic guide can be found in S1 File.

In general, after the introduction of DIALOG+, participants were asked their thoughts on their general impression of the app, including its appeal. This was followed by the main concerns relating to the mental health and wellbeing of adolescents, the relevance of the intervention domains for the adolescent population, the barriers and positive aspects of the app in a medical consultation and the design and layout. Questions on each session were designed to prompt both positive and negative feedback and if needed, further questions were asked in order to obtain additional details on a given opinion. Further details of our experience conducting OFGs is not in the scope of this paper and is discussed elsewhere [19].

## Data collection and analysis

Participants were asked to complete a sociodemographic questionnaire, which included data on gender, age, and educational level for all the participants, prior diagnosis of anxiety and/or depression for the adolescents and years of work experience for the teachers and clinicians. Afterwards, the OFGs were transcribed verbatim and a framework approach was used to identify the main themes affecting the application design, usefulness and functionality. Content analysis was made following the guidelines of Braun and Clark [20]. Thus, a pairwise coding phase was made and afterwards a joint report was written by a social worker and an anthropologist with long experience in qualitative research.

## Results

We conducted 10 OFGs with a total of 45 participants. Throughout all of the OFGs, the majority of participants (77% n = 35/45) were female. Details on the sociodemographic characteristics of participants can be seen in Table 1.

A total of 13 adolescents participated, the mean age of this group was 16 years old and the majority were enrolled in middle school at the time. Mental health problems, in particular anxiety and depression, were investigated in the adolescent population. The majority of participants reported having experienced mental health disorders: 4 reported having both disorders, 4 reported having only anxiety and 4 having neither. Depression without anxiety was only reported by one participant.

**Table 1. Sociodemographic characteristics of participants.**

| Characteristic | Adolescents n = 13 | Clinicians n = 14 | Guardians/Parents n = 6 | Educators/Youth workers n = 12 |
|---|---|---|---|---|
| **Gender** | | | | |
| Male | 5 (38%) | 2 (14%) | 0 (0%) | 3 (25%) |
| Female | 8 (62%) | 12 (86%) | 6 (100%) | 9 (75%) |
| **Age** | 16 (15, 17) | 38 (35, 40) | 48 (43, 52) | 38 (32, 43) |
| **Work experience (years)** | - | 10 (8, 14) | - | 14 (5, 18) |
| **Work experience with adolescents (years)** | - | 4.5 (3.0, 9.8) | - | 10.0 (7.2, 15.2) |
| **Educational level** | | | | |
| Primary school education (grades 1–5) | 0 (0%) | 0 (0%) | 1 (17%) | 0 (0%) |
| Middle school education (grades 6–9) | 8 (62%) | 0 (0%) | 1 (17%) | 0 (0%) |
| High-school education (grades 10–11) | 5 (38%) | 0 (0%) | 2 (33%) | 0 (0%) |
| College education | 0 (0%) | 3 (21%) | 2 (33%) | 5 (42%) |
| Specialization | 0 (0%) | 10 (71%) | 0 (0%) | 4 (33%) |
| Master/PhD | 0 (0%) | 1 (8%) | 0 (0%) | 3 (25%) |
| **Mental health training** | | | | |
| No | - | 9 (64%) | - | 6 (50%) |
| Yes | - | 5 (36%) | - | 6 (50%) |
| **Mental health symptoms** | | | | |
| Both | 4 (31%) | - | - | - |
| Anxiety | 4 (31%) | - | - | - |
| Depression | 1 (7%) | - | - | - |
| Neither | 4 (31%) | - | - | - |

Count (percentage) for categorical variables and median (Interquartile range) for numeric variables.

Information was collected on the time (in years) of working with adolescents for clinicians and educators/youth workers. Educators/youth workers had more experience *(MED = 10, IQR = 7.2, 15.2)*, working with adolescents than clinicians. Clinicians had the highest educational level.

The parents/guardian's group had the highest mean age and the lowest educational level, with only 2 participants with college education.

In order to assess the initial impressions of the intervention, we divided participants views into positive features or appeal (strengths and benefits of DIALOG+) and potential barriers. In the latter, the views of participants on four different categories (privacy and patient-physician relationship, time and connectivity, interface and terminology and additional categories) are shared.

## Positive features

**Innovation and empowerment.** Considering that the use of digital interventions such as DIALOG+ is not common in the Colombian context, we found that the implementation of new technologies in a medical setting was seen as innovative. The incorporation of an app was particularly engaging for our target population.

"*I also feel it is useful and it is very different, like, the use of an application like this during a consultation can be very innovative.*"

(*Adolescent Bogotá*)

All groups agreed that the app allows the discussion of topics that can often be overlooked during a traditional medical consultation. Interaction between patients and clinicians often focuses on physical conditions and exploration of particular aspects of mental health can be neglected. The structured nature of the app was perceived as useful in order to explore these aspects in a direct and detailed manner. Clinicians highlighted that in addition to the order and structure that the intervention might add to the consultation, it may allow them to identify and address specific problems.

"*I thought it was very complete, very interesting, yes, well, generally they never ask you, like the problems you may have or how you are. . .*"

(*Adolescent Duitama*)

"*(. . .) Sometimes during a consultation, we don't have the time to delve into more specific questions about their mental health and we focus on the clinical part of their health, but any individual, particularly a young person has to be evaluated in all aspects. . .*"

(*Clinician Duitama*)

Feelings and emotions can be difficult to identify and verbalize, particularly for adolescents [21]. The approach with key life topics that are explored in the intervention was considered by adolescents as a helpful tool in order to recognize and communicate emotions and difficult situations that they may be going through.

"*(. . .) it can let you recognize the things that you are feeling, because sometimes you have them there, but you don't know what is happening and you can't talk about them and they keep happening but you don't identify them. . .*"

(*Adolescent Bogotá*)

It was considered that the recognition of emotions and deciding over which domains they require additional support, gives adolescents autonomy and allows them to play an active role during their therapeutic process.

"*From the psychologic perspective, it is very interesting to see this tool because it will help [you] identify punctual issues (. . .) it gives the teenager an opportunity of being active in the consultation by expressing what they feel and what they want to improve*"

(*clinician Duitama*)

"[..] *I think it is an attractive alternative, besides it encourages them because I think that through each consultation, they can see in what they have improved [. . .] I think it motivates them, it is striking and can give them concise incentives, normally young people like to challenge themselves.*"

*Parent/guardian Bogotá*

**App navigation.**   In terms of the app interface, all participants agreed that the navigation of the app was easy and intuitive. Moving between screens to rate the different domains and tasks assigned was considered straight-forward. Additionally, the rating scale was perceived as uncomplicated and clear. Participants agreed that these characteristics of navigation should be maintained because they make the app accessible and direct.

"*I think that navigating the app is easy and it is simple enough for someone who is not very familiar with technology—like me- to understand.*"

(*Teacher Duitama*)

## Barriers

**Privacy and patient-physician relationship.**   Both, adolescents and clinicians expressed a concern around privacy during the consultations, as the presence of parents or guardians could prevent adolescents from discussing certain sensitive issues that they might not want their parent/guardian to know about. This was considered important since mental health issues are sensitive and patients need to be comfortable in order to discuss them openly.

Teachers mentioned that this situation may be more of a concern for older teenagers and highlighted that even if parents attend their medical consultations, if the adolescent requests it parents can wait in the waiting room.

Furthermore, adolescents highlighted the importance of building trust with their clinician in order to share sensitive information about their mental health openly and willingly. Most agreed that the discussion of these topics alone enhances trust in the patient-physician relationship, but this process requires time.

Another aspect mentioned, was that the way the intervention is used can change the hierarchical nature of the patient-physician interaction. Some adolescents suggested that they would feel more comfortable using the app by themselves and not having the physician ask the questions, or suggested that the clinician sat beside them and not in front of them while answering the questions.

"*The consultation might be initially uncomfortable, I don't know, well, I think, I have never used anything similar but doing this type of activity requires building some trust and I think*

*the app allows that. (. . .) I would have many questions and the doctor has to be there to answer them but maybe not having him directly looking because that can put a lot of pressure on the patient, but having him close. . .*"

(*Adolescent Bogotá*)

**Time and connectivity.** For clinicians a main concern was time. Since standard consultation time in Colombia is around 20 minutes, they considered that it would be insufficient and could become an impeding factor to correctly apply the intervention during a routine session.

Furthermore, technical issues like availability of devices (tablets, smartphones or computers) and a stable internet connection in health centers, particularly rural settings, were mentioned as major barriers for the broad application of the intervention.

**Interface.** Participants across every OFGs agreed that increasing the interactivity of the app would make it more suitable for the adolescent population as they described it as dull and serious. Suggestions such as allowing personalization of the app according to the age of the participant, including vibrant colors and adding images for each key life topic were given.

"*I think that the graphic interface, like the aesthetics, are a bit dull (. . .) I think in a way It doesn't make you feel comfortable, to put it someway I think it is kind of cold, distant (. . .).*"

(*Adolescent Duitama*)

Clinicians considered that incorporating reminders or incentives with an image or animation when a task was completed could be appropriate.

**Terminology and changes to current categories.** The language and terminology of the app were considered confusing in certain key life topics particularly for adolescents.

The topic "practical help" was confusing for participants with the majority not knowing what it aimed to ask. The Spanish translation for the category "personal safety" was interpreted as personal confidence, instead of the risk of harm they perceive.

Additionally, educators felt that the use of the term "tasks" or "homework" for the actions included for areas where the adolescent indicated they would like more support, was negative and may demotivate young people. Instead, they suggested for it to be changed to a friendlier term like "goals".

Finally, the "mental health" category was considered too wide. Instead, adolescents felt it would be good to have more direct questions asked on this subject to help them give an accurate answer.

"*Perhaps if the questions were more direct in the mental health [category], for example if you are asked "how are you feeling? Have you felt anxious? Concretely (. . .) then it would be easier to answer in the numerical scale, because mental health can go from addictions, to how I woke up feeling in the morning. So, if questions were more direct, they would be easier to answer*"

(*Adolescent Bogotá*)

Since DIALOG+ intervention was developed for use in the adult population, we expected to include additional categories or modify already existing ones in order to make it more appropriate for our target population. With this in mind, the following modifications were suggested:

Initially, since adolescence is a period where the exploration of sexuality and romantic love begins, all participants agreed that separating the category partner/family into two different ones would be ideal. Therefore, the category "partner" would include the discussion of this sensitive topic. Since older adolescents are more likely to have familiarity with this stage, adolescents suggested adding a N/A (not-applicable) answer to the numeric scale. This was considered important not only for categories that may not apply to younger adolescents but in order to give them an option if the adolescent is uncomfortable providing an answer to any particular question.

Throughout the focus groups participants agreed that "job situation" should be changed to "school/academic situation" given that this would be more appropriate for our target population. This subject is viewed as a key aspect in the life of adolescents and that it should be focused on school life in general, such as, academic subjects (grades, teachers), peer relationships and extracurricular activities.

> "*This category should be based on the social aspect, the grades aspect (. . .) well, because I believe that this two are very influential in the emotions of us adolescents (. . .).*"

> (*Adolescent Bogotá*)

However, participants considered that "job situation" should be maintained as a key topic in the intervention. The latter, as well as the topic "housing" were viewed as aspects related to the economic situation of parents or caretakers, and clinicians and guardians highlighted the large impact that these two categories have in the life and stability of any adolescent.

> "*I think that they [adolescents] are very perceptive to the situation of their family, even if it seems that they are not listening, they are very aware of what is discussed at their homes. Like, for example the lack of a job, items that can`t be afforded (. . .) I think that even though it is something they don't express much, they perceive it very well.*"

> (*Clinician Duitama*)

Two categories -practical help and medications- were viewed as important to maintain, particularly for adolescents who may have a chronic condition or disability. However, clinicians and parents/guardians suggested grouping them under a category of "professional accompaniment".

**Additional categories.** Adding five additional key life topics targeted particularly to the adolescent population was suggested by educators, clinicians and guardians. The first one was a "body image" category, where the adolescent's self-perception and the impacts of body image and beauty standards of society in their life can be discussed.

The second one was "life project" where adolescents can discuss what their future achievements and goals are. It was established that this one should be aimed particularly at older adolescents and clinicians emphasized how the prospect of a career and life after school are stressful elements that impact the mental health of this population.

An additional category where the clinician can explore the adolescent's experiences with psychoactive drugs was suggested. However, it was pointed out that discussion of this matter on an initial consultation can make the adolescent apprehensive, particularly if a parent/guardian is in attendance.

> "*We are currently seeing that adolescents are having contact with different substances at an earlier age more frequently: marihuana, cigar, liquor (. . .) we see ten- or eleven-year-old kids*

*that have already had contact with these substances. The consumption of these substances has been normalized."*

(*Teacher Duitama*)

*"(. . .) I don't agree with adding this category because it is not a subject that you can easily discuss on a first visit with an adolescent. I mean, if a kid is consuming and you ask him bluntly, he will say that he doesn't consume, he will get annoyed and won't come back (. . .) It could generate barriers, I think it is better that this comes up during the process with the adolescent"*

(*Teacher Bogotá*)

A category that focuses on religion and spirituality was discussed by clinicians, since they considered that the way adolescents relate to these aspects could in some cases be a protective factor.

Finally, it was suggested that a topic where the use of social networks, video games and internet can be directly assessed could be beneficial. Given that these technological tools have shaped the way we socialize it was considered as an influential factor for adolescents.

## Discussion

The feedback obtained with the OFGs suggest that overall, the intervention and app are seen as acceptable, innovative and have appeal to adolescents as well as adult stakeholders. However, concerns were raised about the availability of time to apply the intervention during a standard consultation, the app interface and the language and terminology used. Additionally, new categories that were considered important to adolescents, such as, school environment and academic performance were suggested.

In line with international organizations, such as, the World Health Organization (WHO) and previous research, this study demonstrates the importance of engaging end-users in the adaptation and development of digital mental health interventions [22]. Overall, the implementation of a patient-centered digital intervention was well received by adolescents, clinicians and stakeholders alike, which is consistent with findings from other studies [23–26]. The incorporation of technology during consultation is particularly appealing to young people who are more likely to be familiar with digital devices and the internet in general [27].

Consistent with the most recent Lancet Commission digital technologies [28], this paper supports the importance of digital health in increasing the access to evidence-based interventions specially in a LMIC setting. In order to make digital interventions widely available in these contexts, it is important to test the use of tools that are already available in different settings and for different age groups. Additionally, in order to make digital interventions relevant a participative process is needed where end users could express their ideas, needs and requirements [26].

The structured nature of DIALOG+ was considered useful by adolescents because it gave them an opportunity to recognize emotions and feelings in an easier way. Also, clinicians felt that they could identify problems in a direct manner with the benefit of knowing the main concerns and priorities of their patients. A key element that is mentioned in the literature regarding technology-mediated interventions in general, is their ability to empower patients by giving them an active role in their mental health, which also makes them more likely to engage in the process [29–31]. Participants considered that DIALOG+ also offered this empowering characteristic because it allows them to express their feelings and what they want to improve.

As expected, however, there are several aspects of the DIALOG+ app and intervention that can be improved and tailored to be more suitable for the adolescent population.

Firstly, the app was considered slightly cold and serious. In order to increase user friendliness and engagement on any app, an appropriate interface and use of language are crucial [9, 21]. Participants suggested modifications that would increase interactivity, such as adding images or symbols to each key life topic, personalization of the app according to age and making it more vibrant and colourful. All of these have been commonly used in other mental health apps [32].

Regarding terminology and language used, participants suggested different modifications. These were considered necessary to clarify the life domains and to avoid misunderstandings. A brief description of what each category is aiming to ask, could be added as a "pop-up" to minimize confusion as suggested by some clinicians. Specifically, we consider that the Spanish translation of "personal safety" could be modified to "environmental safety" (*seguridad en el entorno*) since most participants were confused by its meaning. This modification makes the objective of the category clearer, both for the clinician and the adolescent. Similarly, all participants were especially confused by the category "Practical help". The latter aims to inquire about the resources available to support the patient (government aids, requirement of wheelchair/equipment or any other assistances) [18], but in Colombia these are scarce which makes the category confusing. Therefore, we consider that a modification to "external aids" (*ayudas externas*) could be helpful.

A specific modification discussed by adolescents in our OFGs, was to include a "not-applicable" response to the 1 to 7 grading scale system. This was considered important for categories that may not apply to some adolescents due to their age and for categories where the adolescent does not wish to provide an answer. According to them, a clear option that allows the adolescent to avoid answering any category or question is crucial to recognize their autonomy [14, 29]. Adolescents also underscored the central role of clinicians in making them feel at ease sharing their experiences without feeling judged or lectured. Thus, the confidence in the use of the intervention as well as the clinician's ability to convey trust and comfort is vital.

To apply the DIALOG+ intervention in an efficient but thorough manner, clinicians must be familiar with the app so that they feel at ease using it. However, even if clinicians agreed on the importance of thoroughly knowing the app and all its functions, the time availability to apply the intervention was considered a major barrier for uptake. In the Colombian context medical visits are usually limited to 20 minutes, this was considered insufficient to perform a traditional consult and the DIALOG+ intervention. The discussion of sensitive topics related to mental health is time consuming and addressing them in a fast-paced manner would be counterproductive, particularly for adolescent populations, where the aim is to create trust and improve health outcomes.

The addition and modification of the existing domains may be relevant to adapt the DIALOG+ intervention to the needs of Colombian adolescents. Two main modifications stood out, 1) dividing the "partner/family" domain into two separate ones and, 2) including a school and academic life domain. Both were considered essential by all participants since they relate to key aspects that are particular to the mental health of adolescents. The first one was considered given that adolescence is generally the initial period of sexual exploration and it has been reported that the discussion of sexuality between physicians and adolescents is generally insufficient or lacking [33].

Despite the consensus that for adolescents the domains "Job situation" and "Housing" were strongly related to their parents or guardian's socio-economic status, most of the participants considered that these aspects frequently affect the well-being of any adolescent and thus should remain on the app. It is important to notice that these two domains may have gained relevance after the onset of the COVID-19 pandemic, because of the impact on household income in general and how it has increased poverty and inequality in Colombia [34].

Since pharmacological treatment is seldom used for adolescents with depression and/or anxiety and limited to severe cases where psychosocial interventions were unsuccessful [35], treatment-related categories such as, "practical help" and "medication" would predominantly apply to adolescents with chronic diseases or physical disabilities. We believe that these categories are important to keep, to promote inclusion of a marginalized group, even if these conditions are not highly prevalent during adolescence [36]. These two categories could be grouped under a category of "professional accompaniment" as suggested by participants.

Other aspects of adolescent life were suggested as individual domains. However, some of them can be examined on the already existing domains. For example, the discussion on career projects or future plans that older adolescents experience could be addressed under the domain that includes their academic or school life. Also, aspects regarding the "body image" category and beauty standards, could be included under the "physical health" domain, where questions on body image and self-perception are asked. Finally, the use of social networks and internet as well as the use of substances could be explored when the "friendships" domain is evaluated.

Ultimately, the data obtained provided valuable insight into the areas of DIALOG+ that are susceptible to modification in order to correctly adapt this intervention to the context of Colombian adolescents. Perhaps in the long run the DIALOG intervention could become routinely used in the clinical practice as part of the mental health evaluation of the Colombian youth in order to improve their outcomes, and our experience may facilitate its adaptation into other LMIC countries.

Barriers to the implementation of this type of interventions have to be taken into account since these are often shared by LMIC such as a lack of digital literacy, limited connectivity and access to technology, and availability of time during a consultation in order to properly apply this intervention.

## Limitations

Performing OFG due to the physical distancing required by the COVID-19 pandemic meant that participants not only had to have access to a digital device, but also have a high enough level of computer literacy with a stable internet connection. These characteristics are not predominant in a LMIC like Colombia. Additionally, OFGs were performed with participants from two urban areas of Colombia. Therefore, these findings may not be representative of adolescents, clinicians and stakeholders from rural settings.

## Conclusions

Overall, the use of a technology-based intervention was perceived in a positive manner by all participants, and they considered that it might enhance the relationship between physicians and adolescents in the context of primary health care in Colombia. Modifications of language in order to clarify the objective of the intervention and to assess the adolescent's reality in a more accurate way should be considered. Additionally, making the interface more interactive can make the intervention more approachable and less intimidating to the adolescent. The limitations of time during consultations are a major issue that have to be taken into account as well as the parent or guardian presence during the session.

## Supporting information

**S1 File.**
(DOCX)

## Author Contributions

**Conceptualization:** Carlos Gómez-Restrepo, Laura Ospina-Pinillos, Victoria Jane Bird.

**Data curation:** Carlos Gómez-Restrepo, Arturo Marroquín-Rivera, María Gabriela Calvo-Valderrama, Laura Ospina-Pinillos, Diliniya Stanislaus Sureshkumar.

**Formal analysis:** Carlos Gómez-Restrepo, Arturo Marroquín-Rivera, María Gabriela Calvo-Valderrama, Laura Ospina-Pinillos, Diliniya Stanislaus Sureshkumar, Victoria Jane Bird.

**Funding acquisition:** Carlos Gómez-Restrepo, Victoria Jane Bird.

**Methodology:** Carlos Gómez-Restrepo, Victoria Jane Bird.

**Supervision:** Carlos Gómez-Restrepo, Laura Ospina-Pinillos, Victoria Jane Bird.

**Writing – original draft:** Carlos Gómez-Restrepo, María Gabriela Calvo-Valderrama.

**Writing – review & editing:** Carlos Gómez-Restrepo, Arturo Marroquín-Rivera, María Gabriela Calvo-Valderrama, Laura Ospina-Pinillos, Diliniya Stanislaus Sureshkumar, Victoria Jane Bird.

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
