## [Decision Letter · Decision Letter 0]

25 Apr 2022

PONE-D-22-04627Stakeholder perspectives on a patient-centred intervention (DIALOG+) for improving the mental health of adolescents in Colombia: a qualitative studyPLOS ONE

Dear Dr. Bird,

Thank you for submitting your manuscript to PLOS ONE. After careful consideration, we feel that it has merit but does not fully meet PLOS ONE’s publication criteria as it currently stands. Therefore, we invite you to submit a revised version of the manuscript that addresses the points raised during the review process. My comments are detailed below.

We look forward to receiving your revised manuscript.

Kind regards,

Dylan A Mordaunt

Academic Editor

PLOS ONE

Journal Requirements:

“This work is supported by the Medical Research Council (grant number: MR/S023674/1) Building resilience in adolescence - improving quality of life for adolescents with mental health problems in Colombia (BRiCs study).”

“All authors are supported by the Medical Research Council (grant number: MR/S023674/1) Building resilience in adolescence - improving quality of life for adolescents with mental health problems in Colombia (BRiCs study).

https://mrc.ukri.org/

4. We note you have included a table to which you do not refer in the text of your manuscript. Please ensure that you refer to Table 1 in your text; if accepted, production will need this reference to link the reader to the Table.

Additional Editor Comments:

Thank you for your submission. We have received input from two reviewers. The authors should address all comments by both reviewers. Reviewer 1 has recommended rejection on the basis of a small sample size and lack of statistical analysis. My take on the methods (methodology and epistemology are not described) is that the method is primarily focused on coding to saturation, resulting extraction of themes and using exceprts to illustrate those themes. Synthesis of this data is therefore narrative on the background of the coding. There are quantitative methods that could be applied (e.g. word counts, bag of words, text mining etc) but my take is that this wouldn't be warranted and the question of resulting validity may sit best in the post-publication sphere given the criteria for publication for PLoS One. Ultimately I think the authors should address this their response as to what the epistemology, methodology and rationale for not including statistical analyses are. It's a useful assertion by the reviewer and the authors may add impact to their clinical and policymaker audience by addressing this area of methods.

With specific regards to the criteria for publication:

1. The study presents the results of original research.

2. Results reported have not been published elsewhere.

3. Experiments, statistics, and other analyses are performed to a reasonable technical standard- the main area that needs to be addressed in resubmission is this, as described by both reviewers and expanded on by myself.

4. Conclusions are presented in an appropriate fashion and are supported by the data.

5. The article is presented in an intelligible fashion and is written in standard English.

6. The research meets all applicable standards for the ethics of experimentation and research integrity.

7. The authors should consider whether the article type fits with an Equator checklist such as SRQR or COREQ (qualitative research checklists) and if so, check adherence to these checklists and attach the resulting file as a supplementary file. IF ethics allows, the authors should also consider attaching the transcripts and coding, or even an attachment that includes some of the quotes/data they decided not to include in their text given the value that this may add and that the PLoS format allows for this.

I look forward to receiving your resubmission.

Reviewers' comments:

Reviewer's Responses to Questions

**Comments to the Author**

1. Is the manuscript technically sound, and do the data support the conclusions?

Reviewer #1: No

Reviewer #2: Yes

2. Has the statistical analysis been performed appropriately and rigorously? 

Reviewer #1: N/A

Reviewer #2: N/A

3. Have the authors made all data underlying the findings in their manuscript fully available?

Reviewer #1: No

Reviewer #2: Yes

4. Is the manuscript presented in an intelligible fashion and written in standard English?

Reviewer #1: Yes

Reviewer #2: Yes

5. Review Comments to the Author

Reviewer #1: This is a survey tested DIALOG+ on online focus groups with adults and adolescents (13-16 yo) with current or past depression or anxiety. Although both shared decision-making and internet-based approach are important and timely, the study has substantial concerns.

First, the participants are heterogeneous and the sample size is small.

As the authors wrote, this is a narrative study without any statistical approach. Thus, the sample size is not calculated. However, the sample sizes for each group (n=6~14) are too small to conclude anything. Besides, they have a current or past history of depression or anxiety. Some of them are currently medicated and others are not. This is a too heterogeneous dataset to be published.

Second, this study relies on self-report on anxiety or depression. And the authors conducted only a narrative approach. Where is objectivity? Which piece of information supports the author's conclusions??

Third, as the authors mentioned, this is a qualitative study. There are some possible ways to statistically analyze this kind of dataset. For example, if you structured the OFGs, you can conduct text mining. Why did the authors eschew statistical analyses?

Reviewer #2: I find the paper relevant and adds value to the technology mediated mental health interventions. The authors’ attempt to use multi-stakeholder approach to understand the feasibility, acceptability and suitability of the DIALOG+ interventions in Colombia is appreciated, because given the stigma towards mental illness and the limited availability of mental health services in LMICs, such technology based interventions are crucial. The study team has used appropriate technique to analyze the data.

Below are my comments to strengthen the paper:

• The title of the paper needs modification, as the intervention is focused on treating common mental disorders and not on promoting mental health of adolescents. I would suggest the topic can be reworded as “Stakeholder perspectives on a patient-centered intervention (DIALOG+) for adolescents with common mental disorders in Colombia: a qualitative study”.

• The reason for choosing DIALOG+ interventions is not clearly spelt out in the paper. Highlighting its strengths over other technology enabled interventions would make rationale stronger.

• In the methodology section, whether parental consent was obtained to include adolescents in the study is not reported.

• It is important to mention the language used in conducting OFGs, and the process of translation and transcription.

• Under positive features in the results section, the first sub-theme is innovation and empowerment. In what way DIALOG+ intervention empowers adolescents is not clearly explained with required quotes from participants. This change then need to reflect in the discussion section from line no. 401 to 404.

• I am not sure if it is good idea to include self-esteem domain under physical health category, as explained in lines 470 & 471.

• I notice two important observations that are crucial, one, connectivity and network issues for the technology to run effectively, a major concern in LMICs, particularly in Colombia. Two, clinicians spending more time (more than 20 minutes) in DIALOG+ interventions. I am wondering if any of the stakeholders gave any insights of tackling these two challenges.

6. PLOS authors have the option to publish the peer review history of their article (what does this mean?). If published, this will include your full peer review and any attached files.

Reviewer #1: No

Reviewer #2: No

---

## [Author Response · Author response to Decision Letter 0]

28 Jun 2022

Journal general requirements:

We have ensured that our manuscript meets PLOS ONE's style requirements.

2. We note that you have provided additional information within the Acknowledgements Section that is not currently declared in your Funding Statement. Please note that funding information should not appear in the Acknowledgments section or other areas of your manuscript. We will only publish funding information present in the Funding Statement section of the online submission form. Please remove any funding-related text from the manuscript and let us know how you would like to update your Funding Statement

We have removed the acknowledgements section which contained funding information. Our Funding Statement should read as follows:

“All authors are supported by the Medical Research Council (grant number: MR/S023674/1) Building resilience in adolescence - improving quality of life for adolescents with mental health problems in Colombia (BRiCs study).https://mrc.ukri.org/

3. In your Data Availability statement, you have not specified where the minimal data set underlying the results described in your manuscript can be found. PLOS defines a study's minimal data set as the underlying data used to reach the conclusions drawn in the manuscript and any additional data required to replicate the reported study findings in their entirety. All PLOS journals require that the minimal data set be made fully available. 

Upon re-submitting your revised manuscript, please upload your study’s minimal underlying data set as either Supporting Information files or to a stable, public repository and include the relevant URLs, DOIs, or accession numbers within your revised cover letter. Any potentially identifying patient information must be fully anonymized.

Important: If there are ethical or legal restrictions to sharing your data publicly, please explain these restrictions in detail.

The study’s minimal data set in this case relates to qualitative interviews that were transcribed verbatim, with the original recordings destroyed as per the ethical requirements in each country. It is not possible due to ethical restrictions to make the transcripts available as although the transcripts are anonymised, they may contain potentially identifiable data within the text, which would enable known participants to be identified. We instead have provided quotations to support each of the themes within the analysis. We have stated that “Anyone wishing to access the original qualitative dataset, is required to contact the Colombia PI of the project. A formal written request, outlining the nature of the project is required. Permission to utilise the data would then be sought from the study’s independent advisory board, who provide oversight on all ethical issues relating to the study and liaise directly with the relevant ethical committees.” 

4. We note you have included a table to which you do not refer in the text of your manuscript. Please ensure that you refer to Table 1 in your text; if accepted, production will need this reference to link the reader to the Table.

This has been modified, this table is now table 1 of supplemental information “S1 table 1” (Line 95)

5. Please review your reference list to ensure that it is complete and correct.

No changes were made to our reference list.

Response to reviewer 1:

This is a survey tested DIALOG+ on online focus groups with adults and adolescents (13-16 yo) with current or past depression or anxiety. Although both shared decision-making and internet-based approach are important and timely, the study has substantial concerns.

First, the participants are heterogeneous and the sample size is small.

As the authors wrote, this is a narrative study without any statistical approach. Thus, the sample size is not calculated. However, the sample sizes for each group (n=6~14) are too small to conclude anything. Besides, they have a current or past history of depression or anxiety. Some of them are currently medicated and others are not. This is a too heterogeneous dataset to be published.

Second, this study relies on self-report on anxiety or depression. And the authors conducted only a narrative approach. Where is objectivity? Which piece of information supports the author's conclusions??

Third, as the authors mentioned, this is a qualitative study. There are some possible ways to statistically analyse this kind of dataset. For example, if you structured the OFGs, you can conduct text mining. Why did the authors eschew statistical analyses?

Response 

We thank you for your constructive comments regarding the qualitative methodology employed within the study and have addressed the three points raised in our response below. 

Our results and conclusions are based on the data obtained through online focus groups as well as on the thematic analysis performed on the wealth of data obtained. This analysis is one of the most common forms of analysis in qualitative research. As explained under the data collection and analysis section, every focus group session was audio recorded and transcribed verbatim. Information was analysed and a framework approach was used following six phase guidelines by Braun and Clarke. The main focus was on pairwise coding to saturation, after analysing the different codes and major categories obtained, emerging themes were defined. Afterwards, a joint report with this obtained information, was written by a social worker and an anthropologist, members of the research team, with long experience in qualitative research. This document along with the transcriptions support our conclusions and excerpts of the transcriptions were used in order to illustrate the emerging themes. 

The samples included are appropriate to the methodology, where the numbers per focus group are required to be small enough to allow everyone to express their opinions, but large enough to create a group dynamic. It was not the aim of the paper to compare different groups of individuals, for which we agree a more homogenous sample, and/or quantitative statistics would be required. Rather it was our aim to explore and understand the experience of individuals with anxiety and depression who typically present within healthcare services. 

We consider that the narrative synthesis of the data obtained with the thematic analysis, did not call for additional quantitative methods like text-mining because. It was unlikely that these additional methods would provide new or different information than the one we obtained. 

-----------------

Response to reviewer 2: 

We thank you for the review of our paper and your feedback. We have answered each of your points below. 

Reviewer 2:

General comments

I find the paper relevant and adds value to the technology mediated mental health interventions. The authors’ attempt to use multi-stakeholder approach to understand the feasibility, acceptability and suitability of the DIALOG+ interventions in Colombia is appreciated, because given the stigma towards mental illness and the limited availability of mental health services in LMICs, such technology-based interventions are crucial. The study team has used appropriate technique to analyse the data.

Below are my comments to strengthen the paper:

1. The title of the paper needs modification, as the intervention is focused on treating common mental disorders and not on promoting mental health of adolescents. I would suggest the topic can be reworded as “Stakeholder perspectives on a patient-centred intervention (DIALOG+) for adolescents with common mental disorders in Colombia: a qualitative study”.

Response No. 1

We thank you for your suggestion, indeed the DIALOG+ intervention aims to improve the mental health of adolescents by giving a patient-centred approach and structuring the clinician-patient relationship. Considering this, we have modified the title as suggested describing more accurately the scope of our paper. 

2. The reason for choosing DIALOG+ interventions is not clearly spelt out in the paper. Highlighting its strengths over other technology enabled interventions would make rationale stronger

Response No. 2

We have now included some of the strengths that the DIALOG+ intervention offers, like its interface, the focus on the patients self-reported needs, the structure it gives to a routine clinical consultation as well as a brief description of the positive evidence that it has with adult populations. This can be found in lines 82-89.

3. In the methodology section, whether parental consent was obtained to include adolescents in the study is not reported.

Response No. 3

We apologize for the lack of clarity on this regard; Further explanation on the consent process can now be found under the “procedure” subheading of the methodology section (Lines 142-146). It can be read that informed consent was obtained by parents/guardians as well as assent from adolescents before any data collection began. We have also included that anonymity and the possibility of withdrawal at any given time was explained by researchers since the beginning. 

4. It is important to mention the language used in conducting OFGs, and the process of translation and transcription.

Response No. 4

We have included a brief description on the use of language by facilitators in Lines 154-156. It is explained that the use of language aimed to encourage participation. Additionally, in order to give deeper insight to the facilitation of the focus groups, the topic guide followed on each focus group is included as supporting information file 1. 

The process of transcription is described under data collection and analysis.

5. Under positive features in the results section, the first sub-theme is innovation and empowerment. In what way DIALOG+ intervention empowers adolescents is not clearly explained with required quotes from participants. This change then needs to reflect in the discussion section from line no. 401 to 404.

Response No. 5

We have included a clinician quote where it was perceived that the intervention gives the teenagers an active role because it allows them to freely express their feelings and what they wish to improve (Line 234-236), and a quote by a parent/guardian where the intervention was perceived as encouraging (Line 238-241). Additionally, the previous quote (line 228-230) by an adolescent also highlights that the intervention allows recognition of emotions, related to a feeling of autonomy. The discussion section from line no. 395-397 reflects this addition.

6. I am not sure if it is good idea to include self-esteem domain under physical health category, as explained in lines 470 & 471.

Response No. 6

Participants considered that aspects like body image, societal beauty standards and self-perception could be included in the intervention. We considered that interrogations related to this aspect could be supplementary to those asked in the physical health domain. This domain inquiries about diet and physical activities and linked to these, questions regarding body image and self-perception could be asked. This sub-category was renamed to “body-image”.

7. I notice two important observations that are crucial, one, connectivity and network issues for the technology to run effectively, a major concern in LMICs, particularly in Colombia. Two, clinicians spending more time (more than 20 minutes) in DIALOG+ interventions. I am wondering if any of the stakeholders gave any insights of tackling these two challenges.

Response No. 7

No specific insight on strategies to tackle these two challenges were proposed by stakeholders. These were identified as potential issues, however no spontaneous insight on possible solutions was obtained. These problems clearly require particular attention during the implementation phase and possible solutions require special measures since these are major issues that are faced in the Colombian context.

We thank you again for your time and consideration.

---

## [Editor Report · Decision Letter 1]

13 Jul 2022

Stakeholder perspectives on a patient-centred intervention (DIALOG+) for adolescents with common mental disorders in Colombia: a qualitative study

PONE-D-22-04627R1

Dear Dr. Bird,

We’re pleased to inform you that your manuscript has been judged scientifically suitable for publication and will be formally accepted for publication once it meets all outstanding technical requirements.

Kind regards,

Dylan A Mordaunt, MD, MPH, FRACP

Academic Editor

PLOS ONE

Additional Editor Comments (optional):

Thank you for your resubmission. This now meets the criteria for publication. There are some uses of the present particible in places where past particible should be used, these doesn't change the meaning of the text and is so minor that it would often be missed. I would double-check these in the copy-editing stage.
---

## [Editor Report · Acceptance letter]

3 Aug 2022

PONE-D-22-04627R1 

Stakeholder perspectives on a patient-centred intervention (DIALOG+) for adolescents with common mental disorders in Colombia: a qualitative study 

Dear Dr. Bird:

I'm pleased to inform you that your manuscript has been deemed suitable for publication in PLOS ONE. Congratulations! Your manuscript is now with our production department. 

Kind regards, 

on behalf of

Associate Professor Dylan A Mordaunt 

Academic Editor

PLOS ONE